# Coexistent Detrusor Overactivity-Underactivity in Patients with Pelvic Floor Disorders

**DOI:** 10.3390/healthcare10091720

**Published:** 2022-09-08

**Authors:** Matteo Frigerio, Marta Barba, Giuseppe Marino, Silvia Volontè, Tomaso Melocchi, Desirèe De Vicari, Marco Torella, Stefano Salvatore, Andrea Braga, Maurizio Serati, Stefano Manodoro, Alice Cola

**Affiliations:** 1San Gerardo Hospital, ASST Monza, 20900 Monza, Italy; 2Department of Obstetrics and Gynecology, Milano-Bicocca University, 20900 Monza, Italy; 3Department of Woman, Luigi Vanvitelli University of Campania, 80138 Naples, Italy; 4Obstetrics and Gynaecology Department, Vita-Salute University and IRCCS San Raffaele Hospital, 20133 Milan, Italy; 5EOC-Beata Vergine Hospital, 6850 Mendrisio, Switzerland; 6Del Ponte Hospital, University of Insubria, 21100 Varese, Italy; 7ASST Santi Paolo e Carlo, San Paolo Hospital, 20132 Milano, Italy

**Keywords:** coexistent detrusor overactivity-underactivity, pelvic organ prolapse, underactive bladder, urodynamics, surgery, coexistent overactive-underactive bladder

## Abstract

Introduction and Hypothesis: Pelvic floor disorders represent a series of conditions that share, in part, the same etiological mechanisms, so they tend to be concomitant. Recently, awareness of a new lower urinary tract clinical syndrome has risen, namely the coexisting overactive–underactive bladder (COUB). The etiopathogenetic process, prevalence, and related instrumental findings of COUB are not well-established. We aimed to evaluate the prevalence, clinical features, and urodynamic findings of patients with COUB in a large cohort of patients with pelvic floor disorders. *Methods:* A cohort of 2092 women was retrospectively analyzed. A clinical interview, urogenital examination, and urodynamic assessment were performed by a trained urogynecologist. Based on baseline symptoms, patients were divided into COUB and non-COUB groups, and the degree of concordance between COUB and urodynamic findings, and other parameters related to the clinical aspects of these patients were measured and analyzed. *Results:* 18.8% of patients were classified as COUB. The association between COUB and patients with coexisting detrusor overactivity–underactivity (DOU) was statistically significant and there were substantial similarities in terms of population characteristics, symptoms, and urodynamic findings. *Conclusions:* Our study showed a high prevalence of COUB, and a link between this clinical syndrome and DOU was demonstrated. They showed substantial similarities in terms of clinical and urodynamics correlates. Based on these findings, we do think that urodynamic tests can be useful to improve knowledge on COUB and may be of help in the management of this condition.

## 1. Introduction

Pelvic floor disorders (PFDs) represent a series of conditions—including prolapse, lower urinary tract, bowel, and sexual disorders—usually related to pelvic floor obstetric trauma and quality-of-life impairment [1,2]. Since all these conditions share—at least in part—the same etiological mechanisms, they tend to be concomitant. For instance, patients with pelvic organ prolapse (POP) often complain about lower urinary tract symptoms (LUTS), who tend to recover after prolapse surgery [3,4]. Specifically, LUTS are traditionally divided into storage and voiding symptoms [4,5].

Recently, a new lower urinary tract clinical syndrome, named the coexisting overactive–underactive bladder (COUB), has been identified. This is defined as a syndrome “characterized by coexisting storage and emptying symptoms in the same patient, without implying any specific urodynamic/functional findings or causative physiology; and that these symptoms are suggestive of urodynamically demonstrable coexistent detrusor overactivity/underactivity, but can be caused by other forms of urethro-vesical dysfunctions” [6]. COUB is thought to be multifactorial and involves neurogenic and non-neurogenic factors. Specifically, aging is believed to play a major role due to degeneration and biochemical changes due to cellular damage and apoptosis [6]. However, the etiopathogenetic process is yet to be fully understood, and different hypotheses have been proposed [6,7,8,9,10], including: 

(1) Afferent dysfunctions leading to a decrease or an early start and end of the micturition reflex.

(2) Detrusor rest impairment during the filling phase due to abnormal activation, leading to muscle inefficiency and exhaustion in the voiding phase.

(3) Chronic ischemic-reperfusion injury due to blood supply impairment, involving patchy denervation and substitution of the detrusor muscle with fibrous connective.

(4) Autonomous contraction of small areas of the detrusor (“micromotions”) during the voiding phase due to sparse denervation, inducing detrusor contractions during storage and inefficient activation during the voiding phase.

(5) Bladder outlet obstruction-induced remodeling of the detrusor, involving initial compensatory hypertrophy and later impairment and decompensation [10]. 

To date, the prevalence of COUB and related instrumental findings are not well-established. Specifically, urodynamic tests are not routinely recommended and reserved in the presence of unclear cases or in cases not responding to initial treatment [4]. However, the urodynamic correlates of this syndrome are poorly known, and detrusor overactivity (DO) and/or detrusor underactivity (DU) may or may not be recorded. The definition of clinical and urodynamic patterns of patients with DU and coexisting DO (DOU) may help in the future to establish reliable diagnostic criteria for COUB. This uncertainty led the International Consultation on Incontinence Research Society in 2019 to identify the definition of urodynamic findings of COUB as a research priority. This would be of the utmost importance to better understand the clinical characteristics, natural evolution, and adequate management of COUB syndrome. 

However, to date, there is a paucity of urodynamic studies on the relationship between COUB, and urodynamics diagnoses such as DO and DU. As a consequence, we aimed to evaluate the prevalence, association, clinical features, and urodynamic findings of patients with COUB and/or DOU in a large cohort of patients with pelvic floor disorders.

## 2. Material and Methods

Women who underwent outpatient urodynamics evaluation for PFDs between 2008 and 2016 were retrospectively analyzed. A clinical interview was performed to investigate the presence of LUTS, including overactive bladder syndrome (OAB), urge urinary incontinence (UUI), stress urinary incontinence (SUI), voiding symptoms (VS), and bulging symptoms. All definitions conformed to IUGA/ICS terminology [11]. A gynecological examination was carried out and prolapse-staged according to the POP-Q system. We considered a significant prolapse as any compartment descensus staged ≥ II. The urodynamic assessment was performed including filling cystometry, pressure/flow study, and post-void residual volume (PVR) by a trained clinician, as previously described [12]. All procedures and definitions conformed to the Good Urodynamic Practice Guidelines of the International Continence Society [13]. The following parameters were evaluated during the storage phase: Bladder volume at first desire to void, maximum bladder capacity, presence or absence of leakage with intra-abdominal pressure increasing maneuvers, presence or absence of urgency, presence or absence of leakage with urgency, and maximum filling pressure (PdetMax). The urodynamic observation of involuntary detrusor contractions during the filling phase (spontaneous or provoked) was defined as detrusor overactivity. Urinary leakage during the Valsalva maneuver in the absence of a detrusor contraction was considered urodynamic stress urinary incontinence (USUI), while any leakage associated with detrusor overactivity (DO) was recorded as urge urinary incontinence. During the voiding phase, the following parameters were noted: Maximum flow (Qmax), detrusor pressure at opening (Pdet@op), detrusor pressure at maximal flow(pDet@Qmax), detrusor pressure at closure (Pdet@clo), and postvoid residual (PVR). Voiding dysfunction was defined—according to ICS—as “abnormally slow and/or incomplete micturition, based on abnormally slow urine flow rates and or abnormally high post-void residuals”. Positive PVR (PPVR) volume was defined as a post-micturition residual >100 mL. Detrusor underactivity was evaluated through the Bladder Contractility Index (BCI) (pDet@Qmax + Qmax × 5) proposed by Abrams [14] A BCI < 100 was considered indicative of DU. Based on baseline symptoms, patients were divided into COUB and non-COUB groups. Specifically, in the case of coexisting storage and emptying symptoms in the same patient, she was diagnosed as COUB, otherwise as non-COUB. According to urodynamic findings, the population was divided into DOU and non-DOU. Specifically, in the case of coexisting detrusor overactivity and detrusor underactivity in the same patient, she was diagnosed as DOU, otherwise as non-DOU.

The study obtained local ethics committee approval. Statistical analysis was performed using JMP software version 9.0 (SAS, Cary, NC, USA). Continuous data are reported as mean ± standard deviation, while non-continuous data are shown as the absolute (relative) frequency. The degree of concordance/agreement between COUB and DOU was measured with Cohen’s Kappa [10]. Differences were tested using Student’s *t*-test for continuous parametric data, the Wilcoxon test for continuous nonparametric data, and Pearson’s Chi-squared test for noncontinuous data. A *p*-value < 0.05 was considered statistically significant.

## 3. Results

This study represents a secondary analysis of a previous paper focusing on the agreement of lower urinary tract symptoms and corresponding urodynamic diagnosis [15]. In total, 2092 women with PFDs underwent outpatient urodynamic evaluation in the study period. Full records were available for 1972 of them (5.7% exclusion rate due to partial data). The population characteristics are reported in Table 1. The mean age of the patients was 61.0 ± 12.8 years. Lower urinary tract symptoms and anterior pelvic supports are shown in Table 2. The reasons for performing urodynamic evaluation were stress urinary incontinence in 61.6%, overactive bladder syndrome in 57.5%, and voiding in 35.6% of patients. Bulging symptoms were reported by 42.1% of women, and a significant anterior prolapse (≥2 stage) was found in 43.4% of patients. According to baseline symptoms, 371 (18.8%) patients were classified as having COUB, while the remaining 1601 (81.2%) were classified as non-COUB. Urodynamic findings are reported in Table 3. The most frequent urodynamic findings were voiding dysfunction (50.8%), urodynamic stress urinary incontinence (47.6%), and detrusor overactivity (33.5%). Based on preoperative urodynamic findings, the population was divided into 243 (12.3%) women with DOU and 1729 (87.7%) patients without DOU. The association between COUB and DOU was statistically significant (*p* < 0.001), and the observed proportionate agreement was 75.8%, but agreement according to Cohen κ coefficient (κ = 0.09) resulted in only being slight. The two conditions (COUB and DOU) showed substantial similarities in terms of population characteristics, symptoms, and urodynamic findings, compared to the corresponding controls (non-COUB and non-DOUS). Specifically, age (*p* < 0.001) and menopausal status (*p* < 0.001) were associated with both COUB and DOU (Table 4). Both conditions were related to OAB syndrome, urge incontinence, and voiding symptoms, but inversely related to the presence of SUI (Table 5). Moreover, on a urodynamic basis, both COUB and DOU demonstrated a reproducible footprint, characterized by lower volumes and flow, but higher pressures and postvoid residuals (Table 6). Urodynamic diagnoses demonstrated a higher prevalence of detrusor overactivity, voiding dysfunction, positive postvoid residuals, and a lower bladder contractility index for both conditions. However, while COUB was found to be related to bulging symptoms and significant anterior compartment prolapse, this association was not demonstrated for DOU.

## 4. Discussion

Up-to-date COUB diagnostic criteria have not yet been established, and the urodynamic correlates of this syndrome are poorly understood. Although the definition of COUB suggests a correlation with coexistent DOU, it does not rely on urodynamics tests, which are thus not considered mandatory in the diagnostic workup. This uncertainty is exacerbated by incomplete knowledge about COUB development and the paucity of etiopathogenetic models. Our study showed a high prevalence of both conditions, with 18.8% and 12.3% of women having COUB and DOU, respectively. However, these results were comparable to those previously reported in other populations, such as in women scheduled for prolapse repair [16]. A significant link between the two conditions was demonstrated, with a 75.8% proportion of agreement between COUB and DOU. Moreover, they showed substantial similarities in terms of clinical and urodynamics correlates. 

Specifically, we confirmed the central role of aging in COUB/DOU development, since, in both conditions, age was significantly higher than in respective controls. This is thought to be related to progressive degeneration and biochemical changes caused by cellular damage and apoptosis, thus impairing both neural and/or nonneural functions [6]. This is consistent with previous studies reporting that both male and female patients with DO and concomitant impaired contractility were significantly older than controls [17,18,19]. Estrogen deprivation may also play a role, as suggested by the higher prevalence of menopausal women in both COUB and DOU patients. 

Moreover, as expected, both conditions were associated with a higher prevalence of OAB, urge incontinence, and voiding dysfunction. On the contrary, both COUB and DOU were associated with lower SUI. A possible explanation is that one of the most frequent mechanisms of SUI is found in the decrease in urethral resistance, while one of the proposed mechanisms of COUB and DOU is, on the contrary, represented by bladder outlet obstruction [19,20]. The increase in urethral resistance may induce compensatory hypertrophy of the detrusor at the beginning, which may subsequently evolve into decompensation with progressive deterioration of bladder function. The relationship with bladder outlet obstruction was particularly evident in patients with COUB, in which a significant association was found with both bulging symptoms and anterior compartment prolapse. On the contrary, this was not found in patients with DOU. A possible explanation of this difference is that in patients with neglected prolapse, progressive cumulative detrusor damage is more likely to evolve in isolated DU than coexistent DU and DO. This is consistent with previous papers, reporting up to 40% DU in women with severe prolapse scheduled for surgical repair, but only 6% DOU [16,19].

In our study, urodynamic tests demonstrated substantial similarities in terms of urodynamic findings between the two conditions. In particular, both COUB and DOU showed lower maximum cystometric capacity (MCC) and maximum flow (Qmax), and higher detrusor pressures, including pressure at MCC, opening, Qmax, and closure. Moreover, we recorded a higher prevalence of detrusor overactivity, voiding dysfunction, positive postvoid residuals, and a lower bladder contractility index for both conditions. These findings indicate that COUB and DOU incorporate urodynamic characteristics of both bladder overactivity—such as lower MCC and higher detrusor pressures—and bladder underactivity—such as lower Qmax and higher residuals in the same patient [4,19,21]. In addition to this common urodynamic profile, DOU patients also demonstrated a lower first-desire volume. A possible explanation for the consistent agreement between the two conditions and the substantial similarities in urodynamics findings is that they represent a continuum, with DOU likely representing the most severe form of detrusor abnormalities. This hypothesis is consistent with the fact that patients with DOU presented with additional urodynamic alterations, such as low first-desire volume. This has already been reported by previous studies in which patients with concomitant DU and DO demonstrated lower bladder volumes at first desire to void compared to controls [16,17]. Based on these findings, we do think that urodynamic tests can be useful to improve knowledge about COUB and may be of great help in the management of patients with this condition.

To the best of our knowledge, this is the largest work to evaluate the prevalence, association, and clinical and urodynamic features of COAB and DOU in a cohort of patients with pelvic floor disorders. This may be of great help to better understand this condition. A limitation is the retrospective study design. Future efforts should be addressed to standardize the COUB definition including urodynamic parameters.

## 5. Conclusions

Our study showed a high prevalence of COUB and DOU in women with PFDs, and a significant association between the two conditions was demonstrated. COUB and DOU showed substantial similarities in terms of clinical and urodynamics correlates.

Regarding preoperative urodynamics, continuous data are shown as mean ± standard deviation and non-continuous data are shown as absolute (relative) frequency. BCI = Bladder Contractility Index; MCC = Maximum Cystometric Capacity; pDet@MCC = Detrusor Pressure at Maximum Cystometric Capacity; pDet@Qmax = Detrusor Pressure at Maximum Flow; Qmax = Maximum Flow; PVR = PostVoid Residual; PVR% = PVR/MCC × 100; USUI = Urodynamic Stress Urinary Incontinence.

## Figures and Tables

**Table 1 healthcare-10-01720-t001:** Population characteristics. Continuous data shown as mean ± standard deviation. Non-continuous data shown as absolute frequency (relative frequency).

Age (years)	61.0 ± 12.8
Body Mass Index (kg/m^2^)	26.5 ± 4.7
Parity (n)	1.9 ± 1.2
Instrumental delivery	183 (9.3%)
Maximal birth-weight (g)	3479 ± 702
Menopausal status	1580 (80.1%)

**Table 2 healthcare-10-01720-t002:** Lower urinary tract symptoms and pelvic supports. Data shown as absolute frequency (relative frequency).

Overactive bladder syndrome	1134 (57.5%)
Urge urinary incontinence	790 (40.1%)
Stress urinary incontinence	1215 (61.6%)
Voiding symptoms	703 (35.6%)
Bulging symptoms	817 (42.1%)
Anterior prolapse stage ≥ 2	855 (43.4%)

**Table 3 healthcare-10-01720-t003:** Urodynamic findings. Continuous data shown as mean ± standard deviation. Non-continuous data shown as absolute frequency (relative frequency).

First desire to void (mL)	155 ± 81
Maximum cystometric capacity (mL)	396 ± 99
Opening detrusor pressure (cmH_2_O)	21 ± 14
Maximum flow (mL/s)	19 ± 10
Detrusor pressure at maximum flow (cmH_2_O)	25 ± 18
Closure detrusor pressure (cmH_2_O)	22 ± 16
Urodynamic stress urinary incontinence	939 (47.6%)
Detrusor overactivity	660 (33.5%)
Voiding dysfunction	1002 (50.8%)
Positive post-void residual	280 (14.2%)

**Table 4 healthcare-10-01720-t004:** Population characteristics: COUB versus non-COUB; DOU versus non-DOU. Continuous data shown as mean ± standard deviation. Non-continuous data shown as absolute frequency (relative frequency). Abbreviations: COUB, coexisting overactive–underactive bladder; DOU, detrusor overactivity–underactivity. Statistically significant associations shown in bold.

	COUB			DOU		
	Yes	No	*p* Value	Yes	No	*p* Value
Age (years)	63.4 ± 12.4	60.5 ± 10.8	**<0.001**	66.5 ± 10.8	60.3 ± 12.9	**<0.001**
Body Mass Index (kg/m^2^)	26.3 ± 4.8	26.6 ± 4.7	0.397	27.5 ± 4.7	26.4 ± 4.7	**0.026**
Parity (n)	1.9 ± 1.1	1.9 ± 1.2	0.322	2.1 ± 1.5	1.9 ± 1.1	0.142
Instrumental delivery	35 (9.4%)	148 (9.2%)	0.921	15 (6.2%)	168 (9.7%)	0.096
Maximal birth-weight (g)	3502 ± 692	3477 ± 715	0.831	3493 ± 750	3480 ± 706	0.181
Menopausal status	321 (86.5%)	1259 (78.6%)	**<0.001**	222 (91.4%)	1358 (78.5%)	**<0.001**

**Table 5 healthcare-10-01720-t005:** Lower urinary tract symptoms and pelvic supports: COUB versus non-COUB; DOU versus non-DOU. Continuous data shown as mean ± standard deviation. Non-continuous data shown as absolute frequency (relative frequency). Abbreviations: COUB, coexisting overactive–underactive bladder; DOU, detrusor overactivity–underactivity. Statistically significant associations shown in bold.

	COUB			DOU		
	Yes	No	*p* Value	Yes	No	*p* Value
Overactive bladder syndrome	371 (100%)	763 (47.7%)	**<0.001**	186 (76.5%)	948 (54.8%)	**<0.001**
Urge urinary incontinence	235 (63.3%)	555 (34.7%)	**<0.001**	129 (53.1%)	661 (33.2%)	**<0.001**
Stress urinary incontinence	212 (57.2%)	1003 (62.7%)	**0.049**	128 (52.7%)	1087 (62.9%)	**0.002**
Voiding symptoms	371 (100%)	332 (20.7%)	**<0.001**	103 (42.4%)	600 (34.7%)	**0.019**
Bulging symptoms	199 (53.8%)	618 (39.3%)	**<0.001**	113 (47.5%)	704 (41.3%)	0.071
Anterior prolapse stage ≥ 2	185 (49.9%)	644 (40.2%)	**<0.001**	99 (40.7%)	730 (42.2%)	0.662

**Table 6 healthcare-10-01720-t006:** Urodynamic findings: COUB versus non-COUB; DOU versus non-DOU. Continuous data shown as mean ± standard deviation. Non-continuous data shown as absolute frequency (relative frequency). Abbreviations: COUB, coexisting overactive–underactive bladder; DOU, detrusor overactivity–underactivity. BCI = Bladder Contractility Index; MCC = Maximum Cystometric Capacity; pDet@MCC = Detrusor Pressure at Maximum Cystometric Capacity; pDet@Qmax = Detrusor Pressure at Maximum Flow; Qmax = Maximum Flow; PVR = PostVoid Residual; PVR% = PVR/MCC × 100; USUI = Urodynamic Stress Urinary Incontinence. Statistically significant associations shown in bold.

	COUB			DOU		
	Yes	No	*p* Value	Yes	No	*p* Value
First desire volume (mL)	150 ± 78	156 ± 82	0.168	133 ± 70	158 ± 82	**<0.001**
MCC (mL)	384 ± 94	398 ± 100	**0.005**	352 ± 101	402 ± 98	**<0.001**
pDet@MCC (cmH_2_O)	10 ± 8	9 ± 9	**0.011**	16 ± 12	9 ± 8	**<0.001**
pDet@op (cmH_2_O)	23 ± 13	20 ± 14	**<0.001**	25 ± 14	20 ± 14	**<0.001**
Qmax (mL/s)	16 ± 9	19 ± 10	**<0.001**	10 ± 4	20 ± 10	**<0.001**
pDet@Qmax (cmH_2_O)	26 ± 15	25 ± 18	**0.009**	26 ± 12	25 ± 18	**0.016**
pDet@clo (cmH_2_O)	24 ± 17	22 ± 16	**0.022**	24 ± 15	22 ± 16	**0.002**
PVR (%)	16 ± 26	10 ± 23	**<0.001**	26 ± 30	10 ± 23	**<0.001**
BCI	109 ± 48	121 ± 52	**<0.001**	74 ± 21	125 ± 51	**<0.001**
USUI	161 (43.4%)	778 (48.6%)	0.071	107 (44.0%)	832 (48.1%)	0.232
DO	152 (41.0%)	508 (31.7%)	**<0.001**	243 (100%)	417 (24.1%)	**<0.001**
VD	237 (63.9%)	765 (48.8%)	**<0.001**	181 (74.5%)	821 (47.5%)	**<0.001**
PPVR	74 (20.0%)	206 (12.9%)	**<0.001**	72 (29.6%)	208 (12.0%)	**<0.001**

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
