# Peer review of "Coexistent Detrusor Overactivity-Underactivity in Patients with Pelvic Floor Disorders"

_healthcare, 2022, doi:10.3390/healthcare10091720_

Round 1

Reviewer 1 Report

Thank you for this important work, it will add to the understanding and treatment of female pelvic floor conditions. The over-underactive LUT condition needs the clarification conducted in this study.

This is a secondary analysis of a prior study. As I have not read the prior report, some further help is needed to understand what the two groups are that are being compared:

1.    lower urinary tract clinical syndrome named the coexisting overactive–underactive bladder (COUB). This diagnosis was made without urodynamics – correct?. It would help to outline the criteria used in this diagnosis. For example, for overactivity, the patient reported symptoms of existence of urgency; for underactivity, frequency of voiding and sensations of incomplete voiding.

2.    urodynamics diagnoses as DO and DU or DOU. This diagnosis was made without any reference to method 1, above, correct? It would help to outline the criteria used in this diagnosis. For example, for overactivity, the patient reported symptoms of existence of detrusor contractions at a low filling volume; for underactivity, frequency of voiding and the amount of residual volume needed to diagnose underactivity.

For urodynamic table 3, It would help to state the criteria for each measure in both diagnosis, what were cut-off values (please state in methods, results or table legend) for diagnosis of detrusor overactivity and underactivity. How were measures of opening and closing det pr, for example, used as diagnostic criteria?  This would help to understand the results, analysis and discussion.

DISCUSSION

Thank you for this most interesting discussion that provides greater insight into DOU.

Change   However, this resulted comparable to those previously reported in other populations; to this result is comparable to.

Author Response

REVIEWER 1

Thank you for this important work, it will add to the understanding and treatment of female pelvic floor conditions. The over-underactive LUT condition needs the clarification conducted in this study.

Thank you

This is a secondary analysis of a prior study. As I have not read the prior report, some further help is needed to understand what the two groups are that are being compared:

  1. lower urinary tract clinical syndrome named the coexisting overactive–underactive bladder (COUB). This diagnosis was made without urodynamics – correct?. It would help to outline the criteria used in this diagnosis. For example, for overactivity, the patient reported symptoms of existence of urgency; for underactivity, frequency of voiding and sensations of incomplete voiding.
  2. urodynamics diagnoses as DO and DU or DOU. This diagnosis was made without any reference to method 1, above, correct? It would help to outline the criteria used in this diagnosis. For example, for overactivity, the patient reported symptoms of existence of detrusor contractions at a low filling volume; for underactivity, frequency of voiding and the amount of residual volume needed to diagnose underactivity.

Diagnostic criteria and grouping description was improced in the methids as following

Based on baseline symptoms, patients were divided into COUB and non-COUB groups. 

-the urodynamic observation of involuntary detrusor contractions during the filling phase (spontaneous or provoked) was defined as detrusor overactivity

-Urinary leakage during the Valsalva maneuver in the absence of a detrusor contraction was considered as urodynamic stress urinary incontinence (USUI)

-any leakage associated with detrusor overactivity (DO) was recorded as urge urinary incontinence. 

-Positive PVR (PPVR) volume was defined as a post-micturition residual >100 ml.

-Detrusor underactivity was evaluated through the Bladder Contractility Index (BCI) (pDet@Qmax + Qmax × 5) proposed by Abrams [14] A BCI < 100 was considered indicative of DU. 

Based on baseline symptoms, patients were divided into COUB and non-COUB groups. Specifically, in case of  coexisting storage and emptying symptoms in the same patient, she was diagnosed as COUB, otherwise as non-COUB. According to urodynamic findings, the population was divided into DOU and non-DOU. Specifically, in case of coexisting detrusor overactivity and detrusor underactivity in same patient, she was diagnosed as DOU, otherwise as non-DOU.

For urodynamic table 3, It would help to state the criteria for each measure in both diagnosis, what were cut-off values (please state in methods, results or table legend) for diagnosis of detrusor overactivity and underactivity. How were measures of opening and closing det pr, for example, used as diagnostic criteria?  This would help to understand the results, analysis and discussion.

Edited in methods as following

-the urodynamic observation of involuntary detrusor contractions during the filling phase (spontaneous or provoked) was defined as detrusor overactivity

-Urinary leakage during the Valsalva maneuver in the absence of a detrusor contraction was considered as urodynamic stress urinary incontinence (USUI)

-any leakage associated with detrusor overactivity (DO) was recorded as urge urinary incontinence. 

-Positive PVR (PPVR) volume was defined as a post-micturition residual >100 ml.

-Detrusor underactivity was evaluated through the Bladder Contractility Index (BCI) (pDet@Qmax + Qmax × 5) proposed by Abrams [14] A BCI < 100 was considered indicative of DU. 

DISCUSSION

Thank you for this most interesting discussion that provides greater insight into DOU.

Change   However, this resulted comparable to those previously reported in other populations; to this result is comparable to.

Thank you

Reviewer 2 Report

The authors evaluate the prevalence, association, clinical features, and urodynamic findings of patients with COUB and/or DOU in a cohort of patients with pelvic floor disorders.

The study obtained IRB approval. They must be explaining this acronym.

The study is well designed and the cohort of patients it evaluates is very large, so the data shown are useful for the development of knowledge about COUB and DOU. 

In Table 2 they do not use continuous data, as indicated in the header, so they can remove this explanation in that part of the table.

In Table 3 the first line is in bold, why is that?

In Table 4 there are bold data. They don't say why, they should explain it.

The lines of the tables have different thicknesses. Is there a reason for that?

In the discussion they restate the objectives of the research work, when what they should do is discuss them, as they had already been previously stated.

The authors should revise the last paragraph of the conclusions, as it is not appropriate.

Author Response

REVIEWER 2

The authors evaluate the prevalence, association, clinical features, and urodynamic findings of patients with COUB and/or DOU in a cohort of patients with pelvic floor disorders.

The study obtained IRB approval. They must be explaining this acronym.

Edited as Ethics Committee for clarity

The study is well designed and the cohort of patients it evaluates is very large, so the data shown are useful for the development of knowledge about COUB and DOU. 

Thank you

In Table 2 they do not use continuous data, as indicated in the header, so they can remove this explanation in that part of the table.

Thank you for the suggestion, edited

In Table 3 the first line is in bold, why is that?

That was a typo, edited

In Table 4 there are bold data. They don't say why, they should explain it.

Thank you for the suggestion. Added “In bold statistically significant associations.” for Tables 4,5,6

The lines of the tables have different thicknesses. Is there a reason for that?

There is no specific reason, however we can’t see these different thickness, maybe it is due to system paper formatting. 

In the discussion they restate the objectives of the research work, when what they should do is discuss them, as they had already been previously stated.

Edited as required

The authors should revise the last paragraph of the conclusions, as it is not appropriate.

Removed